# Identification, and Experimental and Bioinformatics Validation of an Immune-Related Prognosis Gene Signature for Low-Grade Glioma Based on mRNAsi

**DOI:** 10.3390/cancers15123238

**Published:** 2023-06-19

**Authors:** Yuan Wang, Shengda Ye, Du Wu, Ziyue Xu, Wei Wei, Faliang Duan, Ming Luo

**Affiliations:** 1Department of Neurosurgery, Wuhan No. 1 Hospital, Wuhan 430061, China; wyuancn@163.com (Y.W.); duanfaliang@126.com (F.D.); 2Brain Research Center, Zhongnan Hospital of Wuhan University, Wuhan 430061, China; yeshengda@whu.edu.cn (S.Y.); wudu2015@whu.edu.cn (D.W.); xzy0727@whu.edu.cn (Z.X.); wei.wei@whu.edu.cn (W.W.)

**Keywords:** immune-related prognostic signature, glioblastoma, bioinformatics, experiment, nomogram, prognosis, WGCNA, mRNAsi

## Abstract

**Simple Summary:**

In this study; four immune-related predictive biomarkers for LGG were identified and proven to be IRGs. The development of more efficient immunotherapy techniques was facilitated by the creation of a prognostic signature to evaluate and forecast the prognosis of LGG patients. The immune-related prognostic score was then used to build a nomogram that was subsequently used to more accurately predict the prognosis of LGG patients

**Abstract:**

Background: Low-grade gliomas (LGGs), which are the second most common intracranial tumor, are diagnosed in seven out of one million people, tending to develop in younger people. Tumor stem cells and immune cells are important in the development of tumorigenesis. However, research on prognostic factors linked to the immune microenvironment and stem cells in LGG patients is limited. We critically need accurate related tools for assessing the risk of LGG patients. Methods: In this study, we aimed to identify immune-related genes (IRGs) in LGG based on the mRNAsi score. We employed differentially expressed gene (DEG) methods and weighted correlation network analysis (WGCNA). The risk signature was then further established using a lasso Cox regression analysis and a multivariate Cox analysis. Next, we used immunohistochemical sections (HPA) and a survival analysis to identify the hub genes. A nomogram was built to assess the prognosis of patients based on their clinical information and risk scores and was validated using a DCA curve, among other methods. Results: Four hub genes were obtained: C3AR1 (HR = 0.98, *p* < 0.001), MSR1 (HR = 1.02, *p* < 0.001), SLC11A1 (HR = 1.01, *p* < 0.01), and IL-10 (HR = 1.01, *p* < 0.001). For LGG patients, we created an immune-related prognostic signature (IPS) based on mRNAsi for estimating risk scores; different risk groups showed significantly different survival rates (*p* = 3.3 × 10^−16^). Then, via an evaluation of the IRG-related signature, we created a nomogram for predicting LGG survival probability. Conclusion: The outcome suggests that, when predicting the prognosis of LGG patients, our nomogram was more effective than the IPS. In this study, four immune-related predictive biomarkers for LGG were identified and proven to be IRGs. Therefore, the development of efficient immunotherapy techniques can be facilitated by the creation of the IPS.

## 1. Introduction

Gliomas account for 31% of primary central nervous system (CNS) cancers. Based on their histological characteristics, the World Health Organization (WHO) classifies gliomas into grades I–IV. Grade IV refers to glioblastomas and grades I–III refer to low-grade gliomas (LGGs) in The Cancer Genome Atlas (TCGA) and other databases. About 10–20% of all primary brain tumors are LGGs, and they generally grow slower than LGGs. People with low-grade gliomas have a median survival time of about 4.7–9.8 years, although some people with subtypes of LGG can live for up to 13 years. The mean incidence of LGGs is 0.7/100,000. Patients diagnosed with LGG are generally young, on average being diagnosed at 21.9 years old for grade I glioma and 33.6 years old for grade II glioma. Despite patients with LGG surviving longer than glioblastoma patients, LGG patients typically undergo aplastic conversion or dedifferentiation into high-grade gliomas (HGG) 4 to 5 years after diagnosis [1,2]. Medical or surgical management of LGG in neuro-oncology can be controversial. Despite this, the aim of therapy is to prolong overall survival (OS) and to maintain a good quality of life. For patients with LGG, treatment decisions are based on certain high-risk signatures. The impact of molecular and tumor biomarkers on treatment will constantly evolve as we improve our understanding of them [3].

Due to the poor prognosis of LGG, new prognostic markers are critically needed. The mRNA stemness index (mRNAsi) based on mRNA expression can accurately quantify the similarity between cancer cells and stem cells and reflect the gene expression characteristics of stem cells. Immunotherapy is widely used in the treatment of various tumors. However, with the heterogeneity of tumors, their intracranial location, and the variety of immunosuppressive tumor microenvironments, immunotherapy has made little advance in LGG during recent decades [4]. Thus, new prognostic biomarkers need to be urgently discovered [5]. Recent advances in bioinformatics have promoted public database mining as a tool for identifying cancer biomarkers. The increasing amount of theoretical research conducted on immunotherapy using immune-related genes (IRGs) has yielded promising results for a range of cancers [6]. Growing evidence shows that immune-related prognostic biomarkers can aid in immunotherapy [7]. Exploring immune-related prognostic biomarkers in prostate cancer, Fu et al. provided guidelines for individualized diagnosis and therapy [8]. However, there is a shortage of research on prognostic indicators in LGG adults: only a few genes have been used for research, and their mechanism of action remains unknown. The treatment of glioma may require more gene targets and binding molecular drugs. Our recently published study, which carefully selected an independent immune-related prognostic marker and provided guidelines for suitable treatments for better outcomes, re-established an innovative IRG-based prognostic model in GBM [9]. Here, we used a similar approach and combined with mRNSsi to construct a prognostic signature of LGG to help clinicians assess patient prognosis, which is the novelty of this study.

## 2. Materials and Methods

### 2.1. LGG and Perineural Tissue Acquisition

From December 2022 to March 2023, 6 LGG tissue samples and 6 normal peritumoral tissue samples were collected from the neurosurgery, Zhongnan Hospital, Wuhan University. All LGG patients underwent pathological diagnosis and signed an informed consent form. The study was approved by the Ethics Committee of Central South Hospital of Wuhan University.

### 2.2. Collecting Immune-Related Genes and Datasets

Figure 1 shows a flow chart of the experiment, which also shows the identification and validation of immune-associated biomarkers based on the difference in mRNAsi scores for LGG prognosis. Standardized fragments per kilobase per million mapped reads (FPKM) of LGG were obtained using the GDC hub of UCSC Xena browser (https://xenabrowser.net/datapages/, accessed on 2 March 2023). However, clinically insufficient tumor samples were excluded as they require further investigation. The present study used a total of 649 LGGs, their clinical variables (stage, grade, age, tumor type, and gender), and their complete survival data. The TCGA-LGG data were normalized via the R package DEseq.2 [10] (log transformation and library-size normalization included).

Additionally, using the Gene Expression Omnibus (GEO) database (http://www.ncbi.nlm.nih.gov/geo/, accessed on 2 March 2023), two independent data cohorts were gathered: GSE43378 [11] and GSE107850 [12]. We started downloading three datasets and performed RMA normalization on them using the R package Affy [13]. Afterward, we matched probe IDs to gene symbols using the annotation files for GPL570 and GPL14951. Through the GlioVis database (http://gliovis.bioinfo.cnio.es/, accessed on 2 March 2023), additional data cohorts (Gravendeel, Rembramdt, and CGGA-LGG) were gathered. MMD1, comprising 364 LGG tissues, was created in order to identify differentially expressed genes (DEGs). Furthermore, MMD2 (n = 649) was formed by merging GSE43378, which includes 51 LGGs with complete survival information, and CGGA, which includes 98 LGGs with complete survival information, to identify prognostic values.

### 2.3. Calculation of the mRNA Stemness Index

A machine learning algorithm for one-class logistic regression (OCLR) was applied to the pluripotent stem cell samples, and the mRNAsi of each LGG sample was calculated in accordance with the methodology of Malta et al. This method was previously used to predict the stemness of cancer [14,15,16]. Then, we divided the TCGA samples into high and low scoring groups according to the median.

### 2.4. WGCNA to Filter Key Module

The IRGs came from the ImmPort database (https://www.import.org, accessed on 3 March 2023). We collected 2498 IRGs from the database and selected 1617 genes that coincided with the MMD2 gene list for further analysis. The 1617 IRGs obtained were constructed into an expression matrix using GoodSamplesGenes and the sample network method. The cut-off value, at Z.Ku 2.5 (Z.ku = (ku − mean(k))/(sqrt(var(k))), also produced exceptional results. The WGCNA R package was subsequently used to create co-expression networks [17]. Then, we used the branch cutting method to divide the IRGs into gene modules [18]. Branches were cut with the crucial parameters set as min-ClusterSize = 30 and deepSplit = 2. By comparing the differences in the module eigengenes, the modules with high correlation were selected according to a criterion of correlation greater than 0.6. As a starting point, we evaluated gene significance (GS) in order to search for key modules connected to the mRNAsi score (selected disease characteristics had high or low mRNAsi scores). Furthermore, the average GS for all genes was used to calculate the module significance (MS). We chose the module that was thought to be the most pertinent and crucial by following the above-described steps.

### 2.5. Determination of Immune-Related Genes with Differential Expression

In order to identify DEGs from low- and high-mRNAsi-score tissues based on MMD1, which comprises 364 LGG samples [19], with the R package “limma”, the expression matrix in MMD1 based on limma was used to identify DEGs with |log2FC| ≥ 0.4 and adjusted *p*-value < 0.01.

### 2.6. Hub Gene Identification

We obtained the results of the WGCNA analysis and used the genes identified as being DEGs as the key genes for inclusion in the subsequent analysis.

### 2.7. Identification of Hub Genes

Using a combined WGCNA and DEG analysis, we discovered all overlapping hub genes, and then, we attempted to eliminate potential prognostic biomarkers. Progression-free survival (PFS), disease-specific survival (DSS), and overall survival (OS) analyses were carried out independently via the R package survival [20]. *p* < 0.05 was set as the threshold. In both survival analyses, genes with significant values were regarded as potential prognostic genes. The R package clusterProfiler [21] functionally annotated the potential prognostic genes for the GO pathway and KEGG enrichment analyses. We then defined the significant KEGG and BP pathway terms with *p* < 0.05 as the criterion.

### 2.8. Chemotherapy Sensitivity Response Predictions

Using the oncoPredict R package, an R package that performs a comprehensive analysis of drug response and drug response markers based on machine learning of cell line data, we calculated the half-maximum inhibitory concentrations (IC50) of some drugs in the LGG sample to predict chemical sensitivity (v0.5).

### 2.9. Translation Level of Hub Gene Expression Identification

The different transcriptional expression levels of hub genes for normal and LGG tissues were validated via the GlioVis database (http://gliovis.bioinfo.cnio.es/, accessed on 2 March 2023) after screening for potential prognostic genes [22]. The Gene Expression Profiling Interactive Analysis (GEPIA) webtool (http://gepia.cancer-pku.cn/, accessed on 3 March 2023) and the GlioVis database were applied to explore the relationships between the hub genes. We also used the Human Protein Atlas (HPA) database (https://www.proteinatlas.org/, accessed on 5 March 2023) to obtain information on the differences in protein expression between normal and LGG samples.

### 2.10. Prognostic Risk System Establishment

After combining the hub genes’ expression levels and the univariate Cox regression analysis of OS’s coefficients (Coef), we used the prognostic genes’ potential prognostic significance to develop an IPS. The following is the definition of the LGG sample risk score (RS):$$\mathrm{Risk}\;\mathrm{score}=\sum_{i=1}^{n}{Coef}_{i}\times {Exp}_{i}$$

The expression level of the hub genes is represented by Exp in the equation, and Coef is the multivariate Cox regression analysis’s regression coefficient. The RSs of LGG in MMD1 and MMD2 can be used to evaluate the risk system’s ability to forecast lethality. Based on the median RS in the above dataset, the group was divided into two (low risk and high risk). Additionally, time-dependent receiver operating characteristic (ROC) (one-, three-, and five-year) curves were created via the “survivalROC” R package [23].

### 2.11. Multivariate Cox Regression Analysis

The RSs based on algorithm and additional important clinical factors (such as family history of primary brain tumors, gender, and age) were selected from the MMD2 data. Then, a univariate Cox analysis of OS was conducted to confirm the system’s prognostic significance. *p* < 0.05 was used as the standard to judge if the risk score and other clinical factors could predict OS in LGG patients. The results were visualized using the R software package forestplot [24].

### 2.12. Nomogram Construction and Verification

We attempted to create a nomogram to better understand the practical application of this risk system. Based on the developed IPS and the R package rms, the nomogram was created after performing a cross-validation to prevent overfitting. The nomogram was then sent through a calibration curve for analysis. Accordingly, the 45° line had the highest degree of prediction potential. In order to investigate the clinical significance of the nomogram, we also used the R package rmda to perform a decision curve analysis [25].

### 2.13. Prognostic Risk Signature Functional Exploration

We employed a gene set enrichment analysis to create a biological behavior predictive risk system. We first calculated the RSs for all samples in the MMD2 data. After that, 2 groups of 649 LGGs were created based on the differences in their hub genes’ expression. After the data were annotated using the c2.cp.kegg.v7.4.symbols.gmt gene set, a gene size (n) of 20% and an FDR of 25% were set as the standards considerably enriched by the KEGG [26] and GSEA (http://software.broadinstitute.org/gsea/index.jsp, accessed on 6 March 2023) pathways, with |ES| > 6, *p* = 0.05.

### 2.14. Relevance between Hub Genes’ Expression and Immune Cells

Immune cells are used as potential independent biomarkers of cancer survival. Therefore, the R program ESTIMATE [27] was used to assess the link between prognostic biomarkers and immunocytes. In order to evaluate the interaction between IPS and immunocytes, we additionally assessed the Immune-related Cell Type relevance through Estimating Relative Subsets of RNA Transcripts (CIBERSORT) (https://cibersort.stanford.edu/, accessed on 7 March 2023) [28].

### 2.15. qPCR

RNA from low-grade glioma tissue and peritumor tissue is first extracted and quantified by a nanophotometer (Implen GmbH, Munich, Germany) using RNAiso Plus (Takara, Kusatsu, Shiga, Japan). The amplification process is monitored in real time using a fluorescent dye, which allows for quantification of RNA initiation. More than 5 biological replicates were designed for each sample tested. HiScript^®^ III RT SuperMix (Vazyme, Nanjing, China) and ChamQ Universal SYBR qPCR Master Mix (Vazyme, Nanjing, China) were used for RT-qPCR with 500 ng total RNA according to the kit’s instructions. GAPDH was used as an internal reference gene, and the 2^−ΔΔCt^ method was used for calculation. Appendix A lists the primers used in our experiment.

### 2.16. Statistical Analysis

All figures and data were compiled in R software (version 4.2.1). The survival curves were plotted using the log-rank test, and differential expression was assessed using the Wilcoxon rank-sum test. Additionally, the continuous variables were examined using Student’s t-test. The statistical significance cut-off was 0.05.

## 3. Results

### 3.1. Key Module Identification

In total, 42 outliers were removed from 607 LGGs for WGCNA (Figure 2A). We assessed adjacency using a soft threshold (β) = 3 (Figure 2B). Afterward, the IRGs were divided into six gene modules and identified as shown in Figure 2C. Gray modules, which were excluded from the following study, were created by grouping genes with low correlation to the relevant characteristic. Disease status was significantly correlated with the remaining five modules (high or low mRNAsi score), and the module that most closely related to disease status was the turquoise module (*p* = 8.4 × 10^−68^, R2 = −0.61). Significant correlations between the blue module’s GS and MM were found (cor = 0.82, *p* = 1.5 × 10^−70^, Figure 2E). GS and MM in the blue, yellow, and brown modules were also meaningful: blue, cor = 0.58, *p* = 1.7 × 10^−14^; yellow, cor = 0.55, *p* = 1.3 × 10^−7^; and brown cor = 0.80, *p* = 2.6 × 10^−33^. As shown in Figure 2I, the turquoise module was the key module because it has the highest MS among the six modules. Figure 2J displays these IRGs’ network heatmaps. A traditional MDS diagram (Figure 2K) demonstrates the independence of the 10 modules.

### 3.2. Hub Gene Screening

We compared the survival curves of the two groups; a Kaplan–Meier analysis suggested that patients with lower mRNAsi scores had worse OS compared with the high-mRNAsi-score group (log-rank *p* = 7.3 × 10^−7^, Figure 3A). In total, 166 DEGs (141 underexpressed and 25 overexpressed) were screened using MMD1 in accordance with the predetermined cut-off criteria (Figure 3B). Additionally, we created a heatmap to display the variations in DEG expression between healthy and tumor tissues (Figure 3C). Finally, combined with the WGCNA results, 30 overlapping hub genes were chosen (Figure 3D).

### 3.3. Screening for Potential Prognostic Genes

We subsequently included the 30 hub genes for OS and DSS analysis; 24 hub genes showed significant OS survival (Figure 3E). In the DSS analysis, 20 genes had *p*-values that were less than 0.05 (Figure 3F), and in the PFS analysis, 18 genes had *p*-values that were less than 0.05 (Figure 3G). Furthermore, a lasso Cox analysis (Figure 3H) was employed to screen the hub genes, which showed significance in terms of survival in the above three analyses. Four prognostic genes were then acquired as prognostic biomarkers for LGG and used in the multivariate Cox regression analysis: C3AR1 (*p* < 0.001, HR = 0.98), IL18 (*p* = 0.009, HR = 1.02), SLC11A1 (*p* = 0.01, HR = 1.02), and MSR1 (*p* < 0.0001, HR = 1.01) (Figure 3I).

Poorer OS (*p* = 0.0045; Figure 4A) and DSS (*p* = 0.0027; Figure 4B) were observed in LGG patients who had higher C3AR1 expressions. The OS and DSS of patients who had higher IL18 expressions were also worse than those with lower IL10 expressions (*p* = 0.0014 and *p* = 0.0001, respectively, in Figure 4C,D). In addition, patients who had higher MSR1 expressions reported poor OS, as shown in Figure 4E (*p* = 0.0001). The results of the DSS analysis of PPP4C matched those of the OS analysis (*p* = 0.00015; Figure 4F). Poorer OS (*p* < 0.0001; Figure 4G) and DSS (*p* < 0.0001; Figure 4H) were observed in LGG patients who had higher SLC11A1 expressions.

### 3.4. Potential Function of DEGs

As shown in Figure 5A, DEGs were included in the GO and KEGG enrichment analysis in order to investigate the roles of these prognostic biomarkers. Additionally, a GO BP analysis indicated that the DEGs were highly enriched in cell activation and migration, the positive regulation of response to external stimulus, myeloid leukocyte activation, leukocyte migration, myeloid leukocyte migration, granulocyte migration, positive regulation of cytokine production, leukocyte activation involved in immune response, leukocyte migration, and the cytokine-mediated signaling pathway. A KEGG analysis indicated that the above genes have a role in cytokine–cytokine receptor interactions, natural-killer-cell-mediated cytotoxicity, the B cell receptor signaling pathway, the chemokine signaling pathway, lipid and atherosclerosis, the TNF signaling pathway, osteoclast differentiation, and the Toll-like receptor signaling pathway (Figure 5B).

### 3.5. Verification of Hub Gene Expression in Normal Tissues and LGG Tissues

We thoroughly compared the expression of four prognostic biomarkers in LGG tissues to normal tissues. C3AR1 (*p* = 8.1 × 10^−5^; Figure 6A), IL18 (*p* = 1.5 × 10^−4^; Figure 6B), MSR1 (*p* = 1.5 × 10^−5^; Figure 6C), and SLC11A1 (*p* = 0.01; Figure 6D) all have higher expressions in LGG than in normal tissues. Additionally, the relationships between prognostic biomarkers were further explored. C3AR1 was strongly related to IL18 (R = 0.825, *p* = 3.3 × 10^−129^; Figure 6E), MSR1 (R = 0771, *p* = 7.6 × 10^−117^; Figure 6F), and SLC11A1 (R = 0.733, *p* = 4.16 × 10^−88^; Figure 6G). As shown in Figure 6H–J, the connection between the other three hub genes is strong. The above results suggest that the prognostic biomarkers may collectively affect LGG prognosis. We verified the translation expression level of all prognostic biomarkers except C3AR1 using the HPA database: SLC11A1, medium staining (Figure 7B); IL18, medium staining (Figure 7D); and MSR1(Figure 7F), medium staining. All prognostic biomarkers showed medium staining. These results indicate that all prognostic biomarkers were evidently expressed in the LGG samples, except for C3AR1, which had no staining in both LGG tissues and normal tissues. The prognostic biomarkers SLC11A1 (Figure 7A), IL18 (Figure 7C), and MSR1 (Figure 7E) were all not detected in the normal samples.

### 3.6. IPS Building

In order to quantify the risk of LGG patients, we then developed a risk-predicting system with all four prognostic biomarkers (C3AR1, IL18, MSR1, and SLC11A1). The following formula was used to compute the risk scores for the LGG samples: Risk score = 0.010 × ExpSLC11A1 + 0.020 × ExpMSR1 + 0.009 × ExpIL18 − 0.022 × ExpC3AR1, that was validated using a multivariate Cox regression analysis (Figure 4F). By setting the median risk score as the criterion, the data from 649 LGG patients in MMD2 were separated into two groups (high-risk-score group, *n* = 324; low-risk-score group, *n* = 325). After performing a survival analysis, we discovered that LGG patients with lower risk scores had better OS than patients who had higher risk scores (*p* = 3.3 × 10^−16^; Figure 8A). Figure 8B displays the ROC values for the risk system (one-year AUC: 0.829; three-year AUC: 0.727; five-year AUC: 0.663). We found that patients in the high-risk-score group had a greater mortality rate than those in the low-risk-score group (Figure 8C). To check the accuracy and reproducibility of this signature, we performed the same analyses as before with MMD2. We continued to divide the LGG patients into high- and low-risk (*n* = 182) groups. Similar to the previous conclusion, the OS of LGG patients with higher risk ratings was considerably worse (*p* = 1.5 × 10^−5^; Figure 8D). MMD2 correctly determined that the expected values at 1, 3, and 5 years would be 0.727, 0.785, and 0.805, respectively (Figure 8E). The results from the TCGA-LGG data are consistent with those in Figure 9F.

### 3.7. Clinical Nomogram Based on Created Immune-Related Prognostic Signature

We tried to build a nomogram to provide doctors a visual prognosis decision-making model. The risk score determined by the IPS and a few crucial clinical factors were first obtained. Risk score (*p* = 2.8 × 10^−6^) and family history of primary brain tumor (*p* = 0.02) were discovered to be significantly associated with the OS of LGG patients in the univariate Cox analysis (Figure 9A). Accordingly, the multivariate Cox analysis showed that the risk score (*p* = 4.4 × 10^−7^) can be used to accurately determine the prognosis of LGG patients (Figure 9B). Next, we developed a nomogram based on the risk score and family history of primary brain tumors (Figure 9C). The calibration curve showed that the nomogram could accurately forecast the survival of LGG patients (Figure 9D–F). The net clinical benefit of the nomogram was also calculated using DCA. The outcome suggests that, when predicting the three-year survival rate, our nomogram was more effective than the IPS (Figure 9G–I). © nomogram and IPS each have their own advantages in various cases.

### 3.8. GSEA Analysis

We carried out a GSEA analysis to explore the potential role of the IPS. Using the standards established before in the KEGG analysis, we came to the conclusion that the IPS influenced MHC class protein complex assembly, MHC Ⅱ class protein complex assembly, and peptide antigen assembly with MHC Ⅱ class protein complexes using GO datasets (Figure 10A,B). The IPS is also effective in identifying asthma, allograft rejection, and autoimmune thyroid disease via KEGG enrichment.

### 3.9. Correlation between Immune Infiltration and IPS in LGG

We used ESTIMATE and CIBERSORT to calculate the degree of immune infiltration of MMD2 and explored their correlation. As indicated in Figure 10C, the risk scores and hub genes both have a strong relation to the ESTIMATE score and immune score. The IPS and immune cell types were strongly related when assessed via CIBERSORT (Figure 10D). These signature and hub genes were significantly related to EMSE (*p* < 0.01), macrophage M1 (*p* < 0.01), mast cell resting (*p* < 0.05), B cells (*p* < 0.01), and NK cell resting (*p* < 0.01).

### 3.10. Predicting Immunotherapy Response

We also assessed some circulated drugs’ IC50. The high-risk-score group had a low IC50 compared to several chemical drugs such as temozolomide (*p* < 0.0001, Figure 11A), staurosporine (*p* < 0.0001, Figure 11B), sepantronium bromide (*p* < 0.001, Figure 11C), dinaciclib (*p* < 0.0001, Figure 11D), dactinomycin (*p* < 0.001, Figure 11E), and bortezomib (*p* < 0.0001, Figure 11F), which means that the above drugs were more sensitive to the high-risk-score group.

### 3.11. Validation of Hub Genes’ Expression

The q-PCR results showed that MSR1 (*p* < 0.05; Figure 12A), IL18 (*p* < 0.05; Figure 12B), SLC11A1 (*p* < 0.05; Figure 12C), and C3AR1 (*p* < 0.05; Figure 12D) were highly expressed in the glioma tissue compared with the normal tissues.

## 4. Discussion

Low-grade gliomas (LGGs), which are the second most common intracranial tumor, are diagnosed in seven out of one million people, tending to develop in younger people. In fact, WHO grade I gliomas are benign and generally occur in children, while tumors of WHO grades II–III are rarely curable and frequently progress to higher grades. Although the survival rate of LGGs is higher, LGGs also cause a reduced quality of life and are life-threatening. Additionally, the development of surgical strategy, radiotherapy, and chemotherapy for LGG has experienced a slowdown. Unfortunately, patients have not experienced significant improvements in their quality of life as a result of the progress made [29]. Thus, effective therapeutic approaches and prognostic markers are urgently needed in clinical treatment. In order to provide new prognostic prediction and immunotherapy for LGG, we therefore sought to identify meaningful prognostic biomarkers.

Immunotherapy is a significant and popular approach in preventing and treating tumors [30]. Presently, increasingly more studies have been exploring new prognostic biomarkers associated with the tumor immune microenvironment, including LGG. Recent research has investigated the microenvironmental landscape in brain tumors, including LGG, HGG, and brain metastases, and uncovered the features of disease-specific immune cells, which has helped us to explore immunotherapy for brain tumors [31]. Tan et al. found a relationship between prognosis for patients with glioma and IRGs via four datasets of GEO and found that the combined use of six candidate genes was effective in assessing the prognosis of glioma, particularly in LGG [32]. However, similar studies on LGG are still rare. In this study, we determined biomarkers associated with the prognosis of LGG. The latest evidence suggests that the mRNA expression-based stemness index (mRNAsi) in LGG has a prognostic value. Zhang et al. developed a stemness-index-based signature with seven genes, having good applications for risk stratification and survival prediction in low-grade gliomas [33]. Thus, we utilized this stemness index to classify LGGs and used several improved methods based on various datasets and databases to explore their immune-related prognostic markers.

We performed DEG and WGCNA analyses based on IRGs. Afterward, 30 overlapping genes were selected from the results. Four IRGs (interleukin 18 (IL18), macrophage scavenger receptor 1 (MSR1), solute carrier family 11 member 1 (SLC11A1), and complement C3a receptor 1 (C3AR1)) were further selected using two different types of survival analyses. Thus, these four IRGs were identified as candidate biomarkers for forecasting the prognosis of LGG and a series of methods were used to further validate them. They were all highly expressed in the high-mRNAsi-score group compared with the low-mRNAsi-score group of LGG according to datasets from GEPIA, TCGA-LGG data, and the HPA database. In previous studies, these four genes all appeared in tumor or other disease immune-related mechanisms. Recent studies have found that pyroptosis, an inflammatory type of cell death sparked by certain inflammasomes, has an influence on the proliferation, invasion, and migration of tumors. IL18 is one of the main activated cytokines in the process of pyroptosis, suggesting a role for IL18 in tumors [34]. The findings of Padala et al. [35] indicated that an analysis of the genotypic and haplotype variants of the IL18 gene play a crucial role in predicting the risk of breast cancer. Previous studies demonstrated that MSR1 exhibited high expressions in M2-like pre-tumor macrophages, which were related to tumorigenesis and development, including immunosuppressive factor generation and angiogenesis [36]. Ji et al. [37] thoroughly analyzed the expression level of MSR1 in LGG and found that LGG tissues expressed significantly more MSR1 than normal brain tissues. Further studies revealed that MSR1 might be involved in changes in the tumor microenvironment (TME) and was a potential prognostic biomarker in LGG. SLC11A1 expresses as a multichannel membrane protein, which is associated with host resistance to some pathogens and iron metabolism. Mutations in SLC11A1 are involved in inflammatory diseases, including Crohn disease and rheumatoid arthritis [38]. A study by Zhu et al. found that rs7573065 in SLC11A1 caused an increased risk and a reduced overall survival rate of prostate cancer, suggesting SLC11A1 as a candidate risk factor and prognostic biomarker for prostate cancer [39]. C3AR1 has been found to have a relationship with prognosis or immune infiltration in a variety of tumors. By affecting the polarization of M2 macrophages, C3AR1 may lead to an immunosuppressive microenvironment, as a result, leading to the progression of esophageal squamous cell carcinoma [40]. In stomach adenocarcinoma (STAD), a high expression of C3AR1 is positively correlated with increased tumor immune infiltration, as well as poor prognosis. C3AR1 can also promote the immune escape of STAD by activating the polarization of M2 macrophages and T cell exhaustion [41].

Overall, this study confirmed that the four IRGs were closely associated with tumor prognosis and could be used as new immune-related prognostic biomarkers for LGG. In addition, through these four prognostic biomarkers, we formed an IPS, which, combined with DEG and WGCNA, showed good clinical applicability. We used the IPS and primary brain family history to create a nomogram, which could be applied clinically to predict OS probability in LGG.

This study had certain limitations. Despite the fact that this study was based on multiple datasets, it lacked further experimental validation. In a follow-up study, more experiments will need to be conducted to clarify the potential molecular mechanism in LGG. We also need to validate the study with more clinical data in the near future.

## 5. Conclusions

In this study, four immune-related predictive biomarkers for LGG were identified and proven to be IRGs. The development of more efficient immunotherapy techniques was facilitated by the creation of a prognostic signature to evaluate and forecast the prognosis of LGG patients. The immune-related prognostic score was then used to build a nomogram and was subsequently used to more accurately predict the prognosis of LGG patients. In summary, creating the immune related model based on mRNAsi is beneficial for clinical doctors to assess the prognosis of patients and determine further treatment.

## Figures and Tables

**Figure 1 cancers-15-03238-f001:**
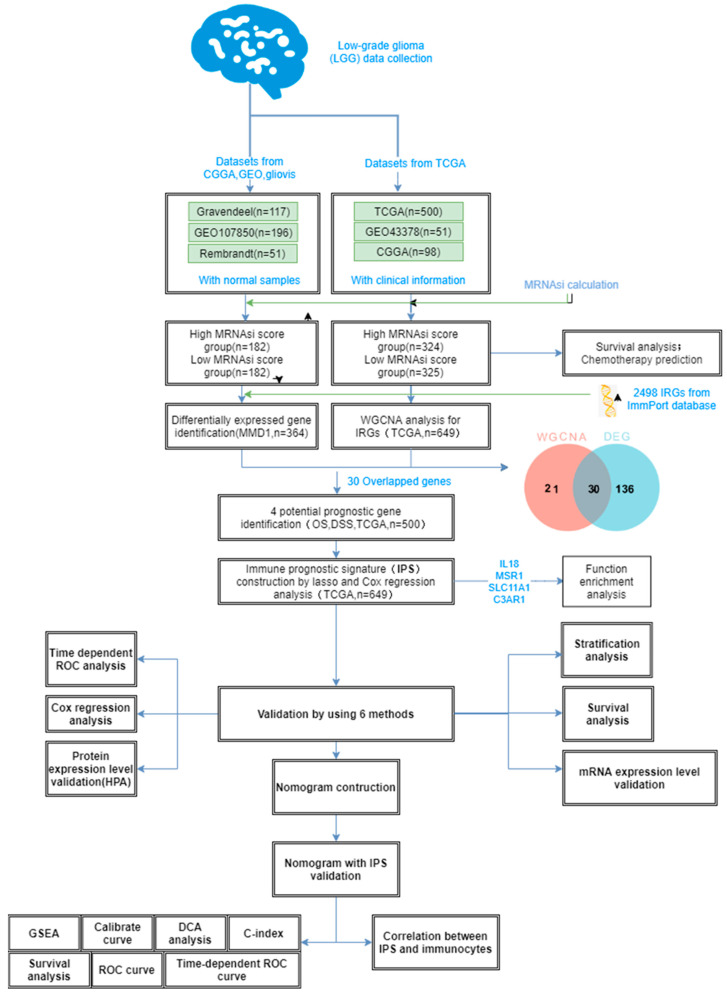
Flowchart for this article.

**Figure 2 cancers-15-03238-f002:**
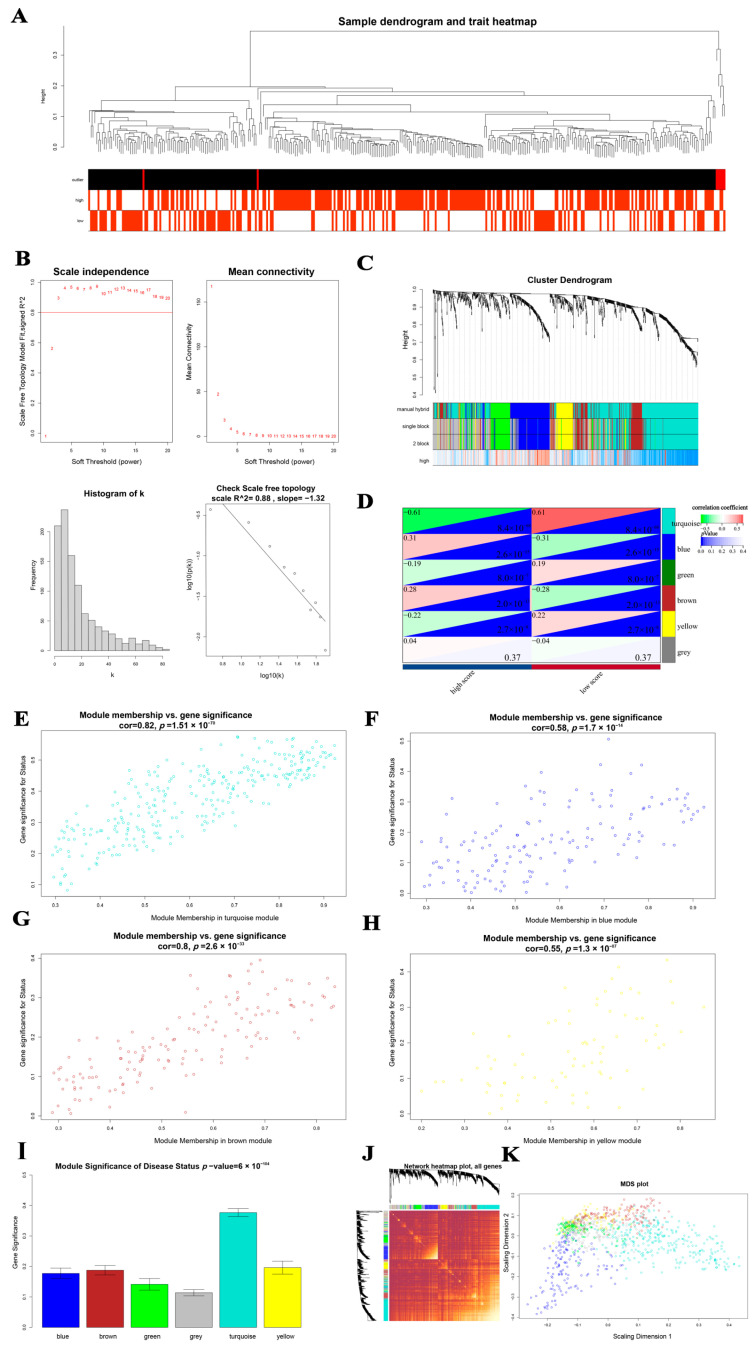
(**A**) A sample dendrogram and trait heatmap. Patient condition is indicated by color intensity. (**B**) Histogram of frequency distribution when = 3; scale-free topology check when = 3. (**C**) Dendrogram of differentially expressed gene groups using the dissimilarity metric (1-TOM). (**D**) Heatmap illustrating the association between clinical data and module eigengenes for LGG. (×10^−H^) (**E**–**H**)Scatter diagrams demonstrating the relationship between module membership and gene significances in blue, black, turquoise, and purple modules. (**I**) The average gene significance distribution in modules associated with the presence of LGG disease. (**J**) Plotting of the network heatmap for all WGCNA genes. (**K**) A conventional MDS plot with the input of TOM dissimilarity. Module assigns a color to each dot with a gene designation.

**Figure 3 cancers-15-03238-f003:**
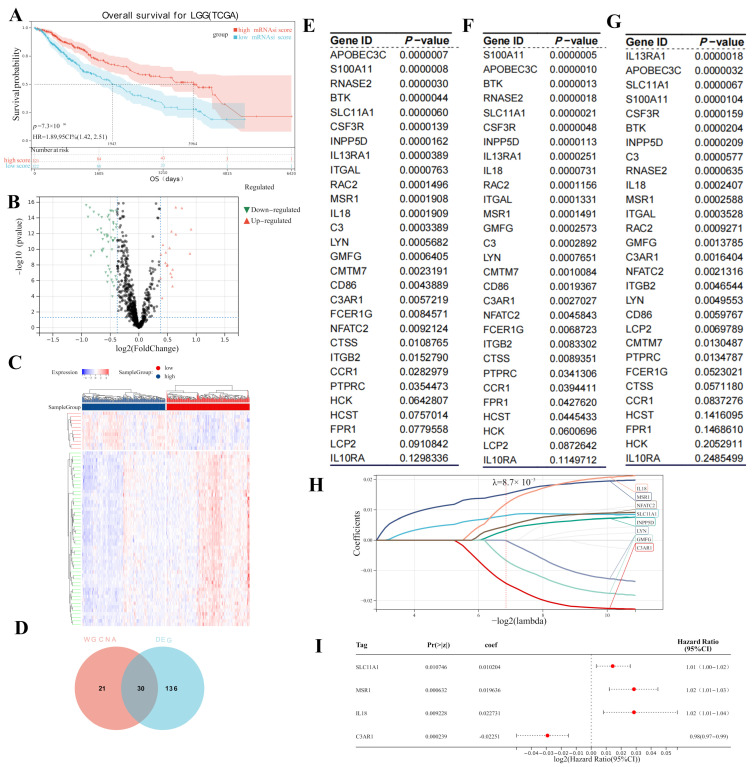
(**A**) Survival analysis in MMD1. (**B**) A volcano plot showing the IRGs that were expressed differently in the MMD1-LGG data. (**C**) Heatmap of IRGs differentially expressed between high- and low-mRNAsi-score samples (fold change > 1, *p* = 0.05, MMD1). (**D**) Venn diagrams showing the common genes found in the DEG and WGCNA. (**E**) Survival analysis (OS) of Venn diagram genes in TCGA-LGG. (**F**) The findings of a separate survival analysis (DSS) for Venn diagram genes in TCGA-LGG. (**G**) The findings of a separate survival analysis (PFS) for Venn diagram genes in TCGA-LGG. (**H**) Lasso Cox curve for hub genes. (**I**) Forest plots of four hub genes.

**Figure 4 cancers-15-03238-f004:**
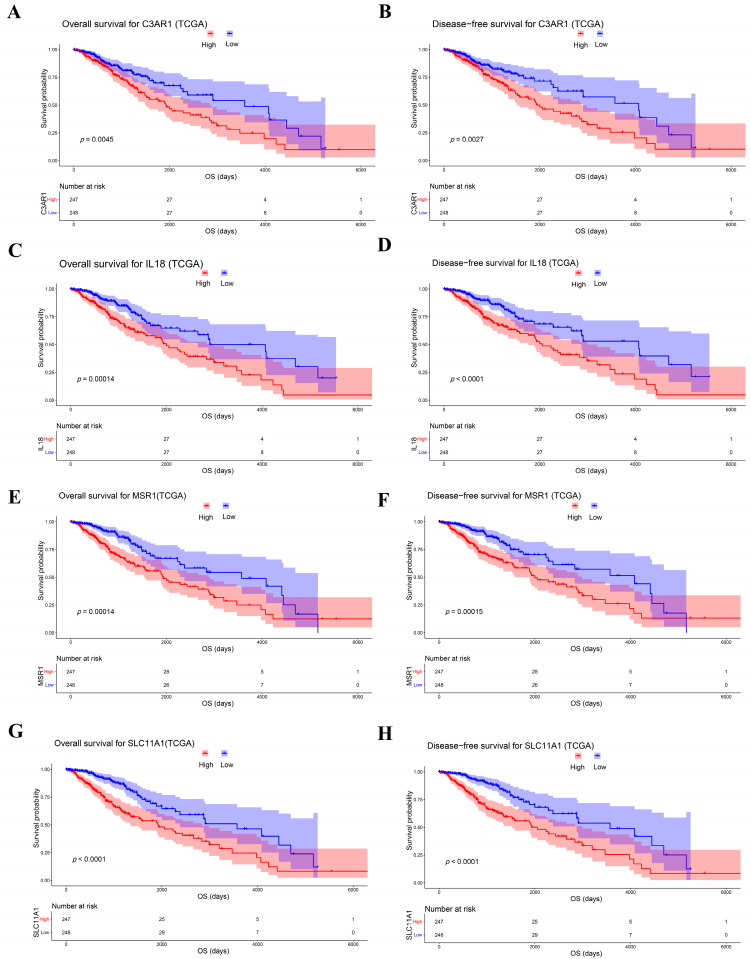
Kaplan–Meier survival curve and disease-free survival curve: (**A**,**B**) C3AR1, (**C**,**D**) IL18, (**E**,**F**) MSR1, and (**G**,**H**) SLC11A1.

**Figure 5 cancers-15-03238-f005:**
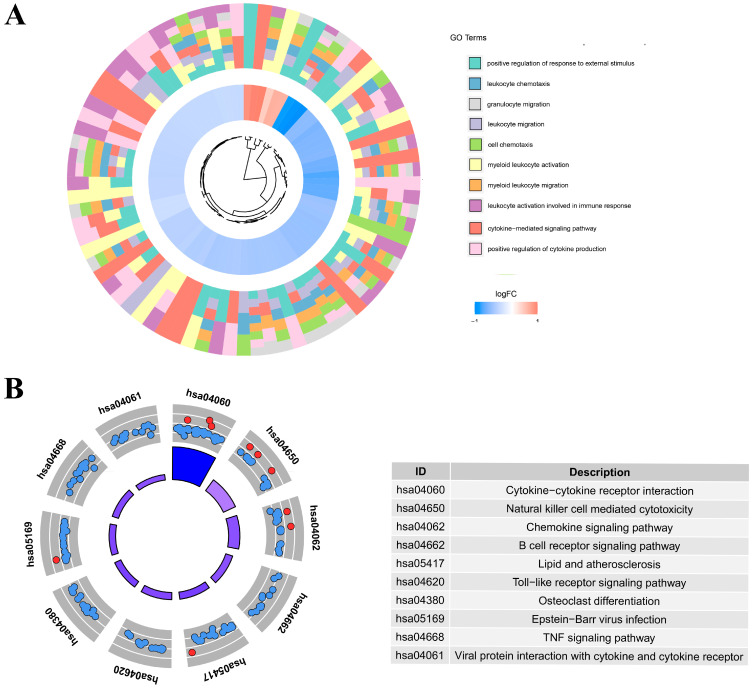
(**A**) Hub IRG GO enrichment plots. The hue of the bars in the inner ring of the bar plot represents the term’s logFC. GO terms are color-coded in the outer ring; (**B**) KEGG circle diagram for IRGs. Bars in the inner circle of the graph are highly responsive to correlation. Red dots represent up-regulated genes, and blue dots represent down-regulated genes.

**Figure 6 cancers-15-03238-f006:**
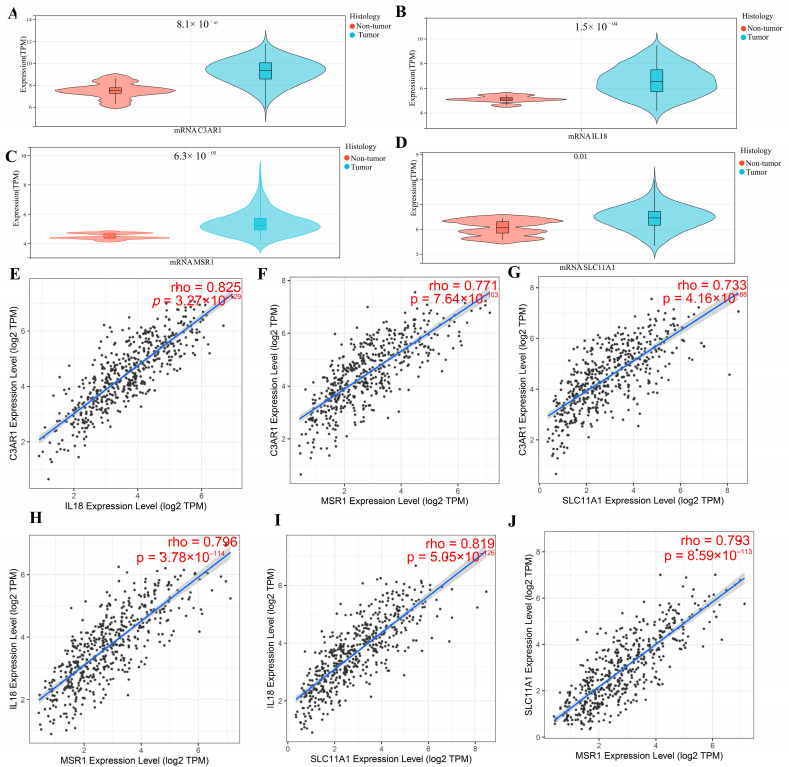
Gene expression levels: (**A**) C3AR1, (**B**) IL18, (**C**) MSR1, and (**D**) SLC11A1 in LGG and in normal tissues. Spearman correlation graph between hub genes: (**E**) C3AR1 and IL-18, (**F**) C3AR1 and MSR1, (**G**) C3AR1 and SLC11A1, (**H**) MSR1 and IL-18, (**I**) SLC11A1 and IL-18, and (**J**) MSR1 and SLC11A1.

**Figure 7 cancers-15-03238-f007:**
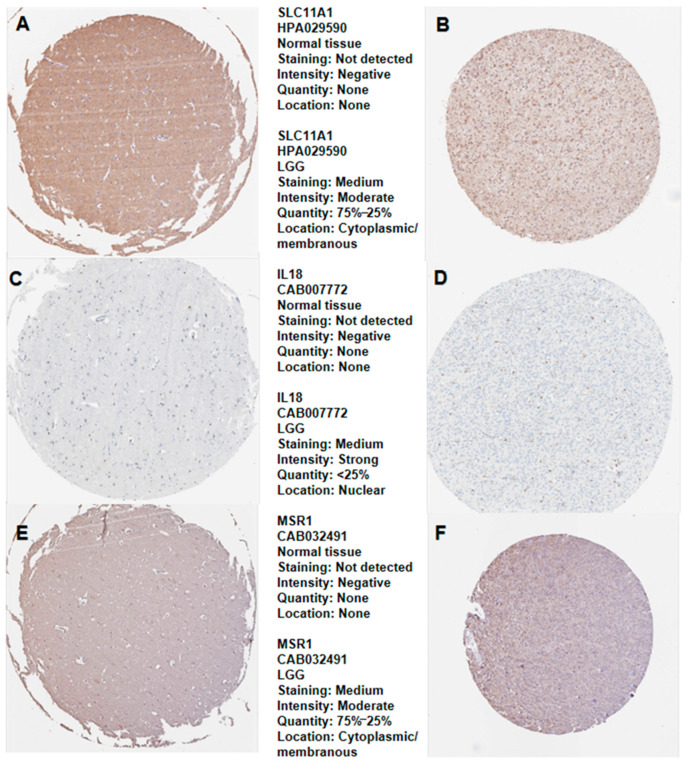
Hub gene expression in normal human and LGG tissue detected by immunochemistry from HPA database. (**A**) Typical IHC staining of SLC11A11 in normal tissue; (**B**) typical IHC staining of SLC11A1 in LGG tissue; (**C**) typical IHC staining of IL18 in normal tissue; (**D**) typical IHC staining of IL18 in LGG tissue; (**E**) typical IHC staining of MSR1 in normal tissue; and (**F**) typical IHC staining for MSR1 in LGG tissue.

**Figure 8 cancers-15-03238-f008:**
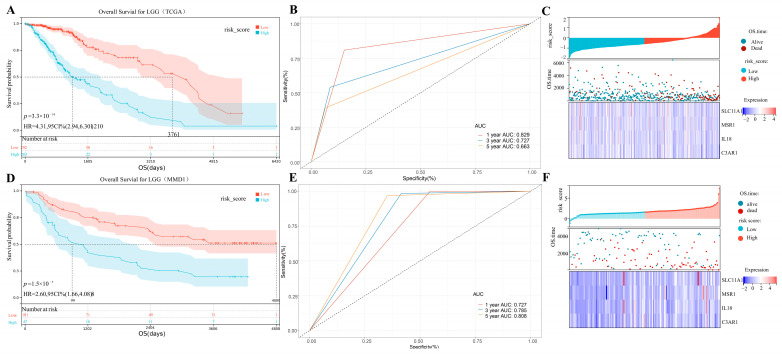
(**A**) Kaplan–Meier OS curves in MMD2. (**B**) Time-dependent ROC for IPS in MMD2. (**C**) Risk assessment heat map based on MMD2: the sample is sorted from lowest to highest risk. (**D**) Kaplan–Meier OS curves in MMD1. (**E**) Time-dependent ROC for IPS in MMD1. (**F**) Risk assessment heat map based on MMD1: the sample is sorted from lowest to highest risk.

**Figure 9 cancers-15-03238-f009:**
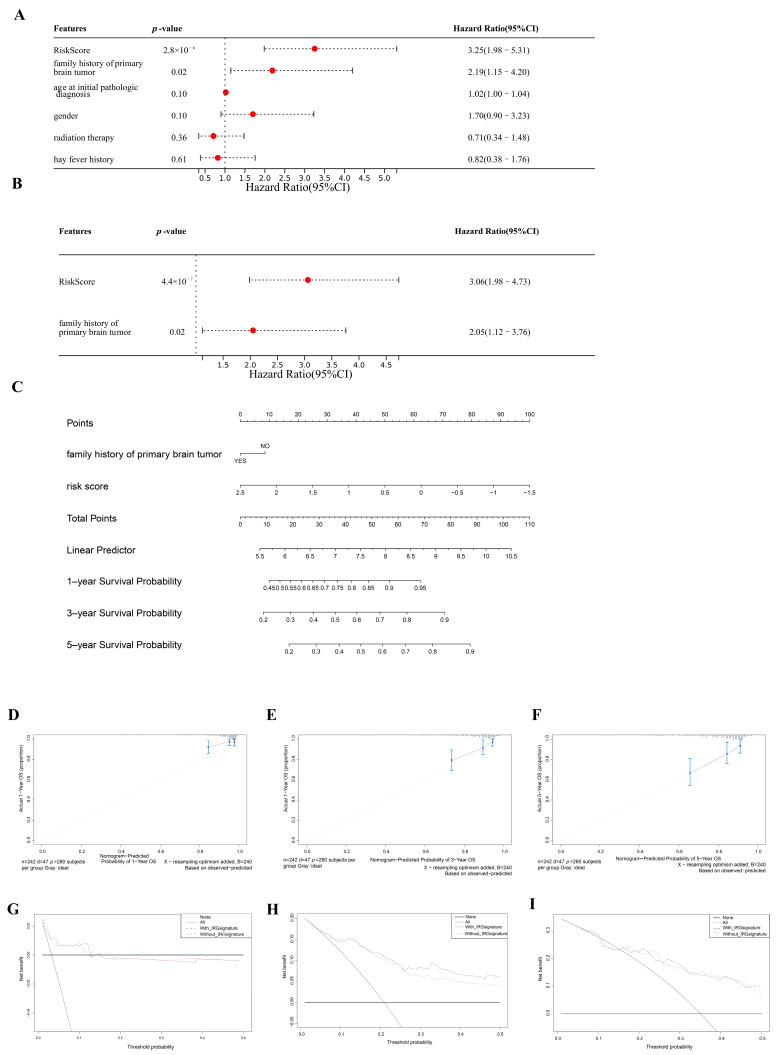
(**A**) The results of the univariate analysis of risk score, family history of brain tumor, age at primary diagnosis, radiation therapy, and gender based on MMD2. (**B**) OS univariate analysis of risk score and family history of brain tumor in MMD2. (**C**) Nomogram for calculating the patients’ one-, three-, or five-year OS. The length of the line indicates the risk level of the factor, and the scale can be used to evaluate the hazard score for each variable. Calibration graphs for forecasting the one-, three-, and five-year OS are shown in (**D**–**F**), respectively. (**G**–**I**) DCA for evaluating the clinical value of immune-related prognostic signatures for one-, three-, and five-year OS; the y-axis displays the net benefit, and the x-axis displays the percent of threshold likelihood.

**Figure 10 cancers-15-03238-f010:**
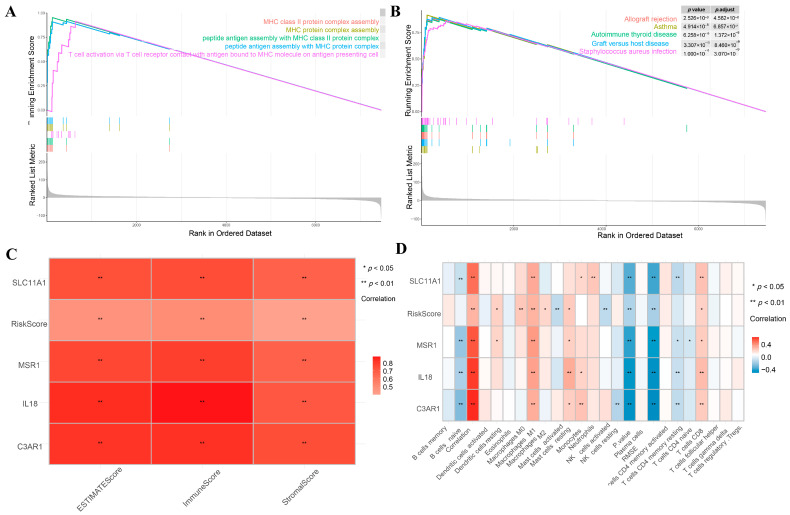
(**A**,**B**) GSEA of IPS. (**C**) Correlations between the risk score, ESTIMATE score, the immune score identified by ESTIMATE. (**D**) Risk score correlations with 22 immune cell activities identified by CIBERSORT.

**Figure 11 cancers-15-03238-f011:**
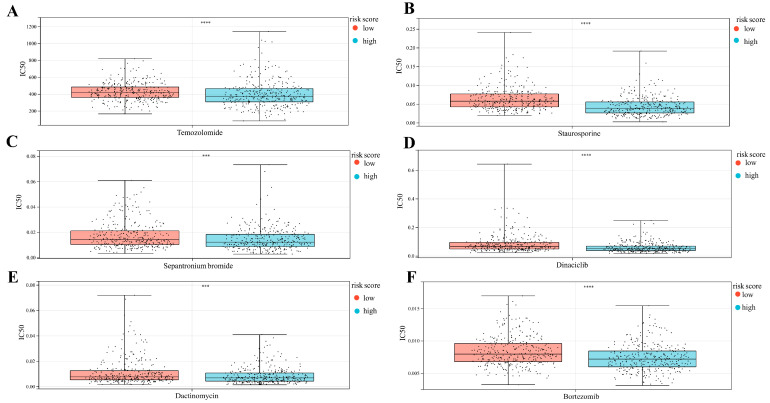
(**A**–**F**) IC50 of six chemotherapy drugs tested using MMD2. (Student’s *t*-test, **** *p* < 0.00001, *** *p* < 0.0001).

**Figure 12 cancers-15-03238-f012:**
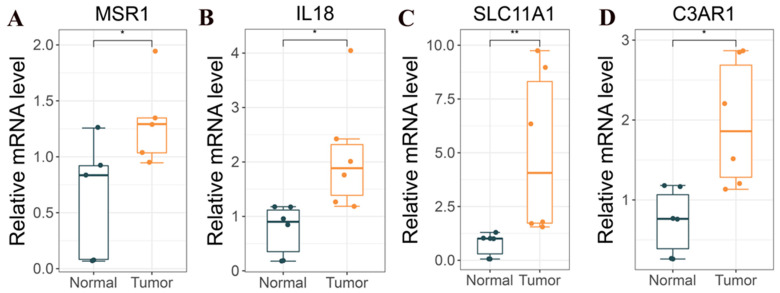
Relative mRNA levels of (**A**) MSR1, (**B**) IL18, (**C**) SLC11A1, and (**D**) C3AR1 in LGG (n = 6) were significantly higher than in normal peritumoral tissues (n = 6) (Student’s *t*-test,** *p* < 0.01, * *p* < 0.05).

## Data Availability

The URLs of all the databases used in this study are provided in the text.

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
