# Peer review of "Identification, and Experimental and Bioinformatics Validation of an Immune-Related Prognosis Gene Signature for Low-Grade Glioma Based on mRNAsi"

_cancers, 2023, doi:10.3390/cancers15123238_

Round 1
Reviewer 1 Report
Wang and co-authors presented a manuscript on the identification of genes useful for immune related prognosis of Low Grade Glioma (LGG).
The manuscript contains many bioinformatic and experimental data that allowed authors to identify four immune-related genes (MSR1, C3AR1, IL-10, and SLC11A1) as predictive biomarkers for LGG. Even if, further experimental works is necessary to validate their finding.
The text contains many typos that need to be corrected, and some sentences are difficult to read. I suggest a revision by a native English to make the text more readable.
In the bibliographic references there are citation numbers repeated twice (3, 9, from 31 onwards). Please correct them.
Almost all figures have very low-resolution graphs and tables and present very small characters which often make them illegible. It is necessary, to make the figures readable and analyzable, to enlarge the characters and increase the image resolution.
Specific points:
- Section 2.14, qPCR: it is described very briefly. It is necessary to indicate which kits, reagents, and instruments were used (with the relative companies). The analysis method, indicated in the last line, should be 2-ΔΔCt, I suppose.
- Figure 6 and related text: which method was used to evaluate the expression of the genes indicated in panel A, B, C, and D? log2TPM, indicated in panel E to G, how was obtained?
- Figure 7. Are the images representing tissue sections with immunohistochemistry? IHC indicate this? This part of the experimental procedure is missing in the materials and methods, and it should be added.
- Figure 12 and related text: SD is missing in the bar graphs; statistical analysis is missing (are the differences statistically significant in all cases?). Which and how many samples were used as normal reference? Which and how many samples were used as tumor tissue? What does mRNA level mean? Is it the RQ corresponding to the 2-ΔΔCt value? The reference sample value, shouldn't it be 1.0?
- Conclusions: 3 immune-related predictive biomarkers are indicated, instead of 4 (as reported in the abstract and along the text).
- Supplementary file. Being described only four couple of primers, I suggest to add this information in Material and Methods, in 2.14 (qPCR) section, along the text.
I'm not a native-English but I found text not fluid to read and in some part not very clear.
I suggest authors a revision by a native English to make the text more readable and to correct the typos present along the text.
Author Response
Dear reviewer,
Thanks very much for taking your time to review this manuscript. I really appreciate all your comments and suggestions! Please find my itemized responses in below and my revisions/corrections in the re-submitted files.
Thanks again!
Comment1: The manuscript contains many bioinformatic and experimental data that allowed authors to identify four immune-related genes (MSR1, C3AR1, IL-10, and SLC11A1) as predictive biomarkers for LGG. Even if, further experimental works is necessary to validate their finding.
Response:First of all, we sincerely thank you for your recognition and valuable comments on this manuscript. In brief, the primary purpose of the present study tends to use multiple bioinformatic methods to reveal the critical role of four immune-related genes as prognostic markers in LGG and further validate the four core gene expression levels using our clinical human samples. We believe that the current dataset provides enough support that four immune-related predictive biomarkers for LGG and the nomogram we created was more effective than IPS. On the other side, you bring up an excellent point that we should consider more experiments in the future and further validate the accuracy of our model and the implication of our findings for developing efficient immunotherapy techniques.
Moreover, considering we only presented 3 hub gene IHC results from the public database in Figure 7, we should show C3AR1 using our own samples. However, we are not able to finish it in the limited review time window (2 weeks). Therefore, we genuinely ask your permission that we could publish this manuscript in its current form, since it is not the core objective of the current manuscript and also because of the time issue. And we will perform more wet-lab experiments as you suggest in our following study.
Comment2: The text contains many typos that need to be corrected, and some sentences are difficult to read. I suggest a revision by a native English to make the text more readable.
Response:We apologize for the poor language of our manuscript. We have now used the MDPI language polishing service and native English speakers helped us with English editing and language corrections. We really hope that the flow and language level have been substantially improved.
Comment3: In the bibliographic references there are citation numbers repeated twice (3, 9, from 31 onwards). Please correct them.
Response: We are truly sorry to ask, which references you are referring to?
Comment4: Almost all figures have very low-resolution graphs and tables and present very small characters which often make them illegible. It is necessary, to make the figures readable and analyzable, to enlarge the characters and increase the image resolution.
Response: Thank you for your feedback. We have rechecked the images in the article, enlarged the font of the text in the image and adjusted the minimum resolution of the images from 300dpi to 600dpi.
Comment5: Section 2.14, qPCR: it is described very briefly. It is necessary to indicate which kits, reagents, and instruments were used (with the relative companies). The analysis method, indicated in the last line, should be 2-ΔΔCt, I suppose.
Response: We are very sorry for our negligence. In the current manuscript, we modified the description of qPCR as follows: RNA from low-grade glioma tissue and peritumor tissue was first extracted and quantified by a nanophotometer (Implen GmbH, Germany) using RNAiso Plus (Takara, Kusatsu, Shiga, Japan). The amplification process is monitored in real time using a fluorescent dye, which allows for quantification of RNA initiation. More than 5 biological replicates were designed for each sample tested.HiScript® III RT SuperMix (Vazyme, Nanjing, China) and ChamQ Universal SYBR qPCR Master Mix (Vazyme, Nanjing, China) were used for RT-qPCR with 500 ng total RNA according to the kit’s instructions. GAPDH was used as an internal reference gene, and the 2-ΔΔCt method was used for calculation. Supplementary Table S1 lists the primers used in our experiment.
Comment6: Figure 6 and related text: which method was used to evaluate the expression of the genes indicated in panel A, B, C, and D? log2TPM, indicated in panel E to G, how was obtained?
Response: As shown in 2.16, the continuous variables were examined using Student’s t-test. And the GlioVis database (http://gliovis.bioinfo.cnio.es/) was applied to explore the relationships between the hub genes indicated in panel 6E to 6G.
Comment7: Figure 7. Are the images representing tissue sections with immunohistochemistry? IHC indicate this? This part of the experimental procedure is missing in the materials and methods, and it should be added.
Response: These images are all from the HPA database and are explained in the methodology. The following explanation is added to the annotation of Figure 7: Hub gene expression in normal human and LGG tissue detected by immunochemistry from HPA database.
Comment8: Figure 12 and related text: SD is missing in the bar graphs; statistical analysis is missing (are the differences statistically significant in all cases?). Which and how many samples were used as normal reference? Which and how many samples were used as tumor tissue? What does mRNA level mean? Is it the RQ corresponding to the 2-ΔΔCt value? The reference sample value, shouldn't it be 1.0?
Response: We thank the reviewer for pointing out this issue. According to your suggestions, we have remade the image and modified the related text as follows: Relative mRNA levels of (A) MSR1, (B) IL18,(C) SLC11A1 and (C) C3AR1 in LGG (n = 6) were significantly higher than in normal peritumoral tissues (n = 6) (Student’s t-test,*P< 0.01, *P < 0.05).
Comment9: Conclusions: 3 immune-related predictive biomarkers are indicated, instead of 4 (as reported in the abstract and along the text).
Response: Thank you very much for pointing out this issue. We should have discovered 4 biomarkers and corrected the incorrect text.
Comment10: Supplementary file. Being described only four couple of primers, I suggest to add this information in Material and Methods, in 2.14 (qPCR) section, along the text.
Response: Thank you for your suggestion, we have added this information along the text.
Reviewer 2 Report
First of all, title of the paper is not correctly phrased. There is extensive need of editing throughout the paper. Figures in the paper are very low resolution figures with extremely small font size making it difficult to understand. Fig. 3A, 3B, 3D what is red and blue label, its very difficult to read. This goes with all the figures in the paper.
Conclusion does not include the significance of the study, and again extensive english editing is required.
Not correct use of grammar. There are lot of repetition of same words in single sentence.
Author Response
Dear reviewer,
Thanks very much for taking your time to review this manuscript. I really appreciate all your comments and suggestions! Please find my itemized responses in below and my revisions/corrections in the re-submitted files.
Thanks again!
Comment1: First of all, title of the paper is not correctly phrased. There is extensive need of editing throughout the paper. Figures in the paper are very low resolution figures with extremely small font size making it difficult to understand. Fig. 3A, 3B, 3D what is red and blue label, its very difficult to read. This goes with all the figures in the paper.
Response: Thanks for the advice, we had the title of the manuscript as Identification, and experimental and bioinformatics validation of an immune-related prognosis gene signature for low-grade glioma based on mRNAsi.
We apologize for the poor language of our manuscript. We have now used the MDPI language polishing service and native English speakers helped us with English editing and language corrections. We really hope that the flow and language level have been substantially improved.
Regarding the figures, we have rechecked the images in the article, enlarged the font of the text in the image and adjusted the minimum resolution of the images from 300dpi to 600dpi.
Comment2: Conclusion does not include the significance of the study, and again extensive english editing is required.
Response: Considering the Reviewer’s suggestion, we have added the following description of significance to the conclusion: In summary, creating the immune related model based on mRNAsi is beneficial for clinical doctors to assess the prognosis of patients and determine further treatment.
Reviewer 3 Report
In this research, several immune-related predictive biomarkers for low grade gliomas were identified and proven to be immune-related genes. The development of more efficient immunotherapy techniques will hopefully be facilitated by the creation of a prognostic signature to evaluate and forecast the prognosis of patients with low grade gliomas. The immune-related prognostic score was used to develop a nomogram that was helpful in predicting the prognosis of patients with low grade gliomas. This is an interesting and somewhat complex paper, although the clinical applications are unclear.
Minor editing would be helpful.
Author Response
Dear reviewer,
Thanks very much for taking your time to review this manuscript. I really appreciate all your comments and suggestions! Please find my itemized responses in below and my revisions/corrections in the re-submitted files.
Thanks again!
Comment1: In this research, several immune-related predictive biomarkers for low grade gliomas were identified and proven to be immune-related genes. The development of more efficient immunotherapy techniques will hopefully be facilitated by the creation of a prognostic signature to evaluate and forecast the prognosis of patients with low grade gliomas. The immune-related prognostic score was used to develop a nomogram that was helpful in predicting the prognosis of patients with low grade gliomas. This is an interesting and somewhat complex paper, although the clinical applications are unclear.
Response: Considering the Reviewer’s suggestion, we have added the following description of clinical significance to the conclusion: In summary, creating the immune related model based on mRNAsi is beneficial for clinical doctors to assess the prognosis of patients and determine further treatment.
And we apologize for the poor language of our manuscript. We have now used the MDPI language polishing service and native English speakers helped us with English editing and language corrections. We really hope that the flow and language level have been substantially improved.
Reviewer 4 Report
The authors provide a gene expression risk score in low-grade glioma with a nomogram that was highly predictive of the prognostic association. The study samples obtained from sources such as TCGA provide a good analysis power. However, the authors could have explained the generation of the mRNAsi score on which they rely on heavily. It is unclear how they incorporate the mRNAsi scores with WCGNA analysis. The details of the generation of the scores and genes included in the high and low groups should be provided. Additionally, the relationship between the sHub IRG (obtained form WCGNA?) and mRNAsi genes should be specified.
Could the author provide a link to genome cancer since it is not available, particularly specifying the study version? For the website, do they mean “Xena”? The authors could specify how many samples from each dataset were used at different stages of the data analysis. For example, were all TCGA, CCGA, and GSE 649 patients used for all analyses? Additionally, these 649 patients should be grouped into the TCGA group only.
mRNAsi should be described in the abstract. For example, how were the IRG genes selected and which ones were up or downregulated? The authors could include the fold change information in Figures 2E,F,G or provide this as a table instead of a figure.
The OS for C3AR1 is different for Cox vs. Kaplan Meyer. In the Cox analysis, the hazard ratio is less than one indicating a better prognosis. Could the authors clarify the discrepancies in this case? Was the cox analysis in 3i adjusted for any stage or other clinical variables?
How were Figure 7 created? Are these from the protein atlas? It should be mentioned in the legend. The authors should rearrange the figures to show which one is normal for which protein. For example, the SLC11A1 is mentioned as normal none, but the staining intensity is high in the top two figures assuming those represent SLC11A1 staining.
What was the benefit of having a lower IC50 for the high-risk group in Figure 10? Did The high-risk group patient show better results with treatment? What were the statistics used in the figure to determine significance?
How were tissues obtained for the qPCR? Unfortunately, there is no clinical and IRB information for these patients, which should be included in the methodology.
Figures should include titles. Larger font sizes can be used for clarity inside the figures.
Author Response
Dear reviewer,
Thanks very much for taking your time to review this manuscript. I really appreciate all your comments and suggestions! Please find my itemized responses in below and my revisions/corrections in the re-submitted files.
Thanks again!
Comment1: he study samples obtained from sources such as TCGA provide a good analysis power. However, the authors could have explained the generation of the mRNAsi score on which they rely on heavily. It is unclear how they incorporate the mRNAsi scores with WCGNA analysis. The details of the generation of the scores and genes included in the high and low groups should be provided. Additionally, the relationship between the sHub IRG (obtained form WCGNA?) and mRNAsi genes should be specified.
Response: Thank you for your question, we have made the corresponding changes in the text, as follows: After we calculated the mRNAsi scores using the method mentioned in 2.3, we divided the TCGA samples into high and low scoring groups according to their median. We further divided the immune-related genes into 6 modules using WCGNA analysis, explored the relationship between each gene module and the mRNAsi score group and identified the closest module for the next analysis.
Comment2: Could the author provide a link to genome cancer since it is not available, particularly specifying the study version? For the website, do they mean “Xena”? The authors could specify how many samples from each dataset were used at different stages of the data analysis. For example, were all TCGA, CCGA, and GSE 649 patients used for all analyses? Additionally, these 649 patients should be grouped into the TCGA group only.
Response: Thank you for your reminder that the Oncomine database is now inaccessible and we have changed the original text as follows: Standardized fragments per kilobase per million mapped reads (FPKM) of LGG were obtained using the GDC hub of UCSC Xena browser (https://xenabrowser.net/datapages/). All TCGA, CCGA, and GSE 649 patients used for all analyses. And we grouped the 649 patients into MMD2. And corresponding modifications have been made to the original text and flowchart.
Comment3: mRNAsi should be described in the abstract. For example, how were the IRG genes selected and which ones were up or downregulated? The authors could include the fold change information in Figures 2E,F,G or provide this as a table instead of a figure.
Response: We have included a description of mRNAsi in the main text and abstract. And we quantified the stem cell characteristics of tumor samples using a machine learning algorithm (OCLR, One Class Linear Regression) to obtain mRNAsi scores and divided the TCGA samples into high and low scoring groups according to their median. Then through DEG and WGCNA, we screened the required IRG genes.
Figures 2E, F, G shows the result of survival analysis (OS, DSS, PFS) of Venn diagram genes in TCGA-LGG rather than the fold change information. We hope to find genes with survival significance under these three indicators for further analysis, so we did not include them.
Comment4: The OS for C3AR1 is different for Cox vs. Kaplan Meyer. In the Cox analysis, the hazard ratio is less than one indicating a better prognosis. Could the authors clarify the discrepancies in this case? Was the cox analysis in 3i adjusted for any stage or other clinical variables?
Response: We re-performed single-factor cox analysis as well as multi-factor cox analysis for these four genes using the survival package. We found that the HR of C3AR1 in the univariate cox analysis was 1.008>1, suggesting that this factor is a risk factor. However, the HR of C3AR1 in the multifactor analysis was 0.981. This may be due to the fact that the expression levels of the four genes were covariate (from Figure 6E-6G, the Spearman correlation curves between the expression levels of C3AR1 and the other three genes show that rho was greater than 0.7), which led to the variation of C3AR1 in the multifactor cox analysis due to the influence of other factors. And the Simpson's paradox was formed.
Comment5: How were Figure 7 created? Are these from the protein atlas? It should be mentioned in the legend. The authors should rearrange the figures to show which one is normal for which protein. For example, the SLC11A1 is mentioned as normal none, but the staining intensity is high in the top two figures assuming those represent SLC11A1 staining.
Response: Figure 7 was from the protein atlas and we had added it in the legend. Andee sort the images horizontally, for example, the SLC11A1 was not detected and the staining intensity is negative in normal tissue (endothelial cells) as shown in Figure 7A. And figure 7B reflects the SLC11A1 staining intensity is middle in tumor tissue.
Comment6: What was the benefit of having a lower IC50 for the high-risk group in Figure 10? Did The high-risk group patient show better results with treatment? What were the statistics used in the figure to determine significance?
Response: The IC50 is the 50% inhibitory concentration, which is the concentration corresponding to the ratio of apoptotic cells to the total number of cells equal to 50%, and can be used to measure the ability of a drug to induce apoptosis; the lower the IC50 value, the more sensitive the high-risk patients are to the drug and the better the treatment effect may be. Considering that IC50 is normally distributed data and comparison between two groups is performed, we used Student's t test to determine the significance of the difference between the two groups.
Comment7: How were tissues obtained for the qPCR? Unfortunately, there is no clinical and IRB information for these patients, which should be included in the methodology.
Response: We have added the following as section 2.1 of the article:
2.1. LGG and perineural tissue acquisition
From December 2022 to March 2023, 6 LGG tissue samples and 6 normal peritumoral tissue samples were collected from the neurosurgery, Zhongnan Hospital, Wuhan University. All LGG patients underwent pathological diagnosis and signed an informed consent form. The study was approved by the Ethics Committee of Central South Hospital of Wuhan University.
Comment8: Figures should include titles. Larger font sizes can be used for clarity inside the figures.
Response: As Reviewer suggested that we added titles to some of the images that were poorly expressed and enriched the description of the picture. We also remade the images and increased the font size of the text in the images to ensure readability.
Round 2
Reviewer 1 Report
The authors partially answered to my comments, but I think that in the present form the manuscript can be accepted.
Reviewer 4 Report
The authors have satisfactorily answered the queries.